# Research on the Response Mechanism of Coal Rock Mass under Stress and Pressure

**DOI:** 10.3390/ma16083235

**Published:** 2023-04-19

**Authors:** Pengfei Shan, Wei Li, Xingping Lai, Shuai Zhang, Xingzhou Chen, Xiaochen Wu

**Affiliations:** 1School of Energy Engineering, Xi’an University of Science and Technology, Xi’an 710054, China; 2Key Laboratory of Western Mines and Hazard Prevention of Ministry of Education, Xi’an University of Science and Technology, Xi’an 710054, China; 3Post-Doctoral Research Workstation, Shaanxi Xiaobaodang Mining Co., Ltd., Yulin 719000, China; 4Bin County Coal Co., Ltd., Xianyang 712000, China

**Keywords:** confining pressure effect, coal rock, stress–strain curve, failure characteristics

## Abstract

In this paper, the strength and deformation failure characteristics of bearing coal rock mass are related to the confining pressure, and the SAS-2000 experimental system is used to carry out uniaxial and 3, 6, and 9 MPa triaxial tests on coal rock to assess the strength and deformation failure characteristics of coal rock under different confining pressure conditions. The results show that the stress–strain curve of coal rock undergoes four evolutionary stages after fracture: compaction, elasticity, plasticity, and rupture. With confining pressure, the peak strength of coal rock increases, and the elastic modulus increases nonlinearly. The coal sample changes more with confining pressure, and the elastic modulus is generally smaller than that of fine sandstone. The stage of evolution under confining pressure constitutes the failure process of coal rock, with the stress of different evolution stages causing various degrees of damage to coal rock. In the initial compaction stage, the unique pore structure of the coal sample makes the confining pressure effect more apparent; the confining pressure makes the bearing capacity of the coal rock plastic stage stronger, the residual strength of the coal sample has a linear relationship with the confining pressure, and the residual strength of the fine sandstone has a nonlinear relationship with the confining pressure. Changing the confining pressure state will cause the two kinds of coal rock samples to change from brittle failure to plastic failure. Different coal rocks under uniaxial compression experience more brittle failure, and the overall degree of crushing is higher. The coal sample in the triaxial state experiences predominantly ductile fracture. The whole is relatively complete after failure as a shear failure occurs. The fine sandstone specimen experiences brittle failure. The degree of failure is low, and the confining pressure’s effect on the coal sample is obvious.

## 1. Introduction

Coal resources are an important part of China’s energy structure. With the development of coal resources, the mining depth has gradually increased, and the surrounding pressure on the coal is increasing. Due to the complex stress environment of deep coal rock, the mechanisms of disaster events during coal rock mining are unclear. The stress state of underground coal rock is complicated and is determined by various factors. Its stress state can be characterized by mechanical analysis; the original rock stress state of the coal rock body is simulated and then studied experimentally, with the coal rock subjected to indoor tests to restore its original mechanical state. The benchmark value of the initial stress state of the coal rock is particularly important, so it is necessary to accurately detect the in situ stress of the research object to provide data to support subsequent research. At the same time, the state of the coal rock is closely related to the safety of the underground construction [1,2,3]. By studying the strength characteristics and deformation characteristics of coal rock under confining pressure, this paper informs understanding of the mechanism of occurrence of coal rock disasters [4,5,6,7,8,9,10,11,12,13,14,15].

With regard to the problem of rock mechanics affected by confining pressure, many scholars have analyzed energy evolution and mechanical properties. Xiaohui Liu [16] et al. analyzed the mechanical properties and failure characteristics of coal rock under different confining pressure conditions and evaluated the influence of confining pressure on coal rock. Cong Cui [17] et al. Carried out research on the mechanical characteristics of coal rock under the condition of a true triaxial equivalent confining pressure and determined the strength and failure characteristics of coal rock. Taoli Xiao [18] et al. studied the strength, elastic modulus, and deformation modulus of deeply buried Dali and showed that, under the action of confining pressure, the elastic modulus and characteristic point strength increased with an increase in confining pressure. Zhizhen Zhang [19] et al. determined the relationship between elastic energy and stress in energy distribution from the perspective of rock energy evolution under different confining pressures and concluded that the confining pressure could change the efficiency of energy accumulation. Hong Wang [20] studied the mechanical properties and constitutive relationship of sandstone in the Yu Heng mining area under different confining pressure, observing that, with changes in the confining pressure, the confining pressure effect on sandstone at each evolutionary stage was different. Fei Ren [21] studied coal rock’s mechanical characteristics and failure modes under triaxial compression and analyzed the failure morphology, elastic modulus, and compressive strength under different confining pressures. Li Cui [22] et al. studied the confinement pressure effect of coal rock from the perspective of the coal rock damage deformation law and acoustic emission characteristic analysis and identified a law of coal rock damage deformation under different confining pressure conditions. Quansheng Liu [23] et al. studied coal rock under high stress under different confining pressures; the confinement effect of coal rock failure after the peak was obvious and showed ductility characteristics. Moreover, the higher the confining pressure, the more obvious the observed ductility characteristics are. Peisen Zhang [24] et al. conducted seepage tests on sandstone with different confining pressure and cyclic load and confirmed the five evolution stages of sandstone, including the primary microfracture compression closure stage, the elastic compression stage, the crack stability development stage, the crack rapid development stage, and the post-peak deformation failure stage. The authors pointed out that rock cracking stress and damage stress are important indicators for determining the rock failure process. The above research greatly enriched our understanding of the mechanical coal rock mass’s mechanical properties and deformation characteristics in confining pressure conditions. It has been confirmed that the environment of deep coal resources is very complex; the occurrences of coal rock in this challenging environment are subject to high confining pressure.

This paper seeks to inform understanding of the mechanisms of occurrence of coal rock disasters. Experiments were conducted on coal rock under the same confining pressure state to obtain the strength and deformation characteristics of different coal rock types under the same confining pressure state, and to determine the relationship between the confining pressure and the elastic modulus, peak strain, coal rock failure mode, and characteristic points of coal rock at each stage. By changing the confining pressure state of coal rock, simulating the underground confining pressure environment of coal rock, and analyzing the whole process of the confinement pressure effect of bearing coal rock, it can be seen that the distribution and strain proportions of the characteristic parameters of coal rock changed according to the confining pressure. The distribution of the characteristic parameters and the regional strain ratio changes of the coal-bearing coal rock samples throughout the process were analyzed; The confining pressure effect of the coal rock mass was elaborated in detail, which provided information on certain specific parameters relevant to impact pressure research and roadway support.

## 2. Experimental Design and Conditions

Considering the roadway layout in the construction process of deep underground engineering facilities typical of coal resource mining and utilization, and due to the need to satisfy the principle of parallel to the maximum horizontal principal stress, we have combined actual geological environment data with the second level of main stress focus MPa to study the influence of confining pressure changes on mechanical properties. It is divided according to the equal gradient, with the confining pressure of 0 MPa, 3 MPa, 6 MPa, and 9 MPa used, respectively, to carry out a triaxial compression test.

### 2.1. Sample Preparation

As shown in Figure 1, we used a standard coal rock sample. In the laboratory, the specimen is prepared under the test procedure of the International Society of Rock Mechanics. The model is made into a cylinder of ϕ50 mm (diameter) × 100 mm (height). To ensure the consistency of the physical and mechanical properties of the coal rock sample and avoid the influence of the initial random fracture of the coal rock sample on the test results, before the start of the test, the coal rock samples with apparent cracks on the surface were first removed through observation with the naked eye.

The naming rules for coal rock specimens are as follows: uniaxial compressive strength (UCS): coal and rock are distinguished from C and R; 0, 3, 6, and 9 represent the confining pressure, respectively; triaxial compressive strength (TCS): coal and rock are distinguished from C and R; 0, 3, 6 and 9 represent the confining pressure, respectively. To avoid accidental phenomena in the discrete distribution of data caused by the anisotropy of the sample, three tests are performed for each state, and the test results are averaged.

Secondly, with the help of an RSM-SY5 (T) non-metallic acoustic wave detector, the initial damage degree of coal rock samples was depicted using the longitudinal wave velocity index, and coal samples with similar longitudinal wave velocities were screened to ensure that the homogeneity met the conditions.

### 2.2. Design of Experiments

The uniaxial state experimental group was added as a blank group to study the mechanical response mechanism of bearing coal rock mass under different confining pressure conditions. First, after the press indenture balances through the test ball, the coal rock is placed in the middle of the upper and lower pressure blocks, the LVDT ring extensometers are installed around the specimen, the positioning rod and the fixed disc are used to constrain and improve in both directions to ensure that the specimen end face is horizontal, the LVDT axial extensometer is set, and then the uniaxial compression test specimen installation step follows. The uniaxial compression test is started; the servo pressure system, the supporting oil cooling system, and the system number set to zero are the physical parameter inputs to be monitored of the coal rock sample. During the test, to facilitate the analysis of the time consistency of the monitoring data, in the reloading stage, the force value control (0.5 KN/s) method is used to apply a 2 KN preload. After the stress value reaches the preload, the axial displacement, axial strain, nor radial strain is cleared again. In the formal loading stage, To obtain a complete stress–strain curve, the axial constant displacement control loading method is adopted, and the operation of the press is controlled according to the rate of 0.1 mm/min. The press starts to run until the specimen is destroyed and the uniaxial compression test ends.

Before starting the triaxial compression test, the spherical balance indenture installed in the original single-axis compression state must be removed. During the installation of the specimen, according to the requirements of the false triaxial test, a high-performance heat shrinkable tube (blocking the high-pressure oil source) is used to wrap the coal rock sample, and the upper and lower pressure blocks as a whole are heated with a high-temperature hair dryer to make it completely fit the surface of the coal rock sample to ensure that the coal rock sample is uniformly stressed during the test. LVDT axial and ring extensometers are installed around the specimen. Then, the coal rock specimen is placed in the middle of the upper and lower pressure blocks, the triaxial compression test is assembled, and the upper end of the pressure chamber is lowered with the help of the pressure chamber lifting device, which is anastomosed with the bottom sealing rigid pressure plate, and the triaxial section is tightly closed. Then, the oil return valve of the confining pressure system is closed, the valve of the external interface of the pressure chamber is opened, and the steady flow filling pump is started to fill it with exhaust oil; when the hydraulic oil seepage of the valve outside the external interface of the pressure chamber occurs, the steady flow filling pump is stopped, the valve of the external interface of the pressure chamber is closed, which ends the preliminary preparation for the false triaxial compression test.

The triaxial compression test starts. The servo pressure system is created first, and the system number and physical parameter input monitoring parameters are cleared for the coal rock sample. During the test, to facilitate the analysis of the time consistency of the monitoring data, in the preloading stage, the force value control (0.5 KN/s) method is used to apply 2 KN preload, and after the stress value reaches the preload, the shaft displacement, axial strain, and radial strain are cleared again. In the formal loading stage, the confining and axial pressures are begun, and the confining pressure is kept constant after reaching the predetermined value (3 Mpa, 6 Mpa, and 9 MPa). Values such as shaft displacement, axial strain, and radial strain are then cleared to zero. The axial constant displacement control loading method is adopted in the formal loading stage to obtain a complete stress–strain curve. The press is started at a rate of 0.1 mm/min until the coal rock sample is destroyed and the triaxial compression test is completed.

### 2.3. Experimental System

As shown in Figure 2, the research group developed a customized SAS-2000 multi-field-coupled rock mass dynamic disturbance triaxial rheological experimental system (Changchun Xin te Testing Machine Co., Ltd., Sinter Company, Changchun, China). According to its own needs, it tested coal rock samples’ uniaxial and triaxial compressive strength.

## 3. Analysis of Coal Rock Mechanical Response Results

### 3.1. Analysis of Basic Mechanical Parameters of Bearing Coal Rock Samples

The response of the basic mechanical parameters of coal rock samples under different confining pressure parameters is shown in Table 1 and Table 2.

For coal samples, the average peak strength was 15.870 MPa in the uniaxial compression state. In contrast, in the three-axis compression state, with the increase in confining pressure, the moderate peak intensity at 3 MPa, 6 MPa, and 9 MPa was 36.357 MPa, 68.230 MPa, and 95.459 MPa, respectively, which increased by 129.09%, 329.92%, and 501.49% compared with the uniaxial compressive strength. In the uniaxial compression state, the average peak strain becomes 0.0128, while in the three-axis compression state, with the increase in confining pressure, the moderate peak strain at 3 MPa, 6 MPa, and 9 MPa is 0.0226, 0.0379, and 0.0396, respectively, which increases by 77.00%, 196.66%, and 209.44% compared with the uniaxial peak strain, respectively. In the uniaxial compression state, the average Young’s modulus was 1.633 GP. In contrast, in the three-axis compression state, with the increase in confining pressure from 3 MPa to 6 MPa, and then to 9 MPa, the average Young’s modulus was 1.817 GPa, 1.944 GPa, and 2.863 GPa, respectively, which increased by 11.27%, 19.02%, and 75.31% compared with the single-axis Young’s modulus, respectively.

Obviously, for the bearing coal rock, confining pressure effectively improves its mechanical properties. The peak strength, peak strain, and Young’s modulus all show an apparent positive correlation with the confining pressure, indicating that the confining pressure can significantly change the mechanical properties of the bearing coal sample. At the same time, combined with data analysis, it can be obtained that the confining pressure is susceptible to the mechanical response of the coal sample, which is reflected in the increment of the data, which is closely related to the initial natural random fracture of the coal sample. Therefore, analyzing the confining pressure mechanism of bearing coal samples is significant.

For fine sandstone, the average peak strength is 82.797 MPa in the uniaxial compression state. In contrast, in the three-axis compression state, with the increase in confining pressure, the average peak strength at 3 MPa, 6 Mpa, and 9 MPa is 122.677 MPa, 135.715 MPa, and 157.500 MPa, respectively, which increases by 48.17%, 63.91%, and 90.23% compared with the uniaxial compressive strength, respectively. In the uniaxial compression state, the average peak strain becomes 0.0091, while in the three-axis compression state, with the increase in confining pressure, the moderate peak strain at 3 MPa, 6 MPa, and 9 MPa is 0.0122, 0.0116, and 0.0115, respectively, which increases by 33.87%, 27.29%, and 25.82% compared with the uniaxial peak strain. In the uniaxial compression state, the average Young’s modulus was 11.113 GP. In contrast, in the three-axis compression state, with the increase in confining pressure from 3 MPa to 6 MPa, and then to 9 MPa, the average Young’s modulus was 11.143 GPa, 12.559 GPa, and 15.485 GPa, respectively, which increased by 0.26%, 13.01%, and 39.34% compared with the single-axis Young’s modulus, respectively.

For fine sandstone, the mechanical properties are also improved due to confining pressure. However, due to the typical strong brittleness characteristics of the rock samples, the initial fractures are few, so the positive correlation changes are mainly reflected in the peak intensity and Young’s modulus. In contrast, the peak strain improvement mechanism is insignificant, showing a trend of rising first and then remaining unchanged. Data comparison shows that the sensitivity of the mechanical data of the bearing rock sample is far less than that of the coal sample, which indirectly indicates the essential difference between coal and rock.

### 3.2. Analysis of Stress–Strain Curve in the Whole Process of Bearing Coal Rock Samples

Figure 3 and Figure 4 show the stress–strain curve of bearing coal rock samples under different confining pressure conditions. Using the indoor rock mechanics data to draw the stress–strain curves of the whole process of using coal rock samples under different confining pressure conditions, it can be found that the presence of confining pressure significantly increases the bearing capacity of the pieces with two other properties of coal rock, so the peak strength, elastic modulus, and residual strength are positively correlated with the confining pressure conditions. However, they show differences due to the different natural initial damage of coal and rock (coal joint fracture development and rock joint fracture development).

Table 3 is a statistical table of residual strength parameters of coal rock samples; according to the test results, the pre-peak stage of the stress–strain curve in the whole process of bearing coal rock samples has high similarity, and there are the compaction stage, elastic stage, plastic stage, and post-peak stage. In the compaction stage, the original microfractures of the coal rock specimen are gradually closed, the deformation amount is large, the bearing capacity is steadily enhanced, the confining pressure is increased, and the compaction stage becomes shorter or even negligible, which is due to the fact that the primary fracture has been compacted during the confining pressure. In the elastic step, the elastic deformation of coal rock specimens is obvious; the stress–strain curve is close to a straight line. In this stage, mainly recoverable deformation accumulates a large amount of elastic energy. With prominent elastic deformation characteristics and the increase in confining pressure, the flexible set is extended, and the stage stores more energy to complete the accumulation of the limit load. In the plastic stage, the coal rock specimen begins to move away from the trend of the elastic stage phenomenon curve, and there may also be a small range of stress reduction and fluctuation; the stage of irreversible deformation then occurs. The formation of irreversible damage becomes aggravated; the coal rock specimen begins to show plastic deformation failure, the bearing capacity of the model reaches its limit, and the stress reaches its peak. With the increase in confining pressure, the plastic stage is also extended, and its entire failure process increases. In the post-rupture location, the specimen rupture is more prominent; the stress decreases rapidly after the peak strength but does not drop to zero. The specimen still has a specific bearing capacity; there is residual force, and the model in the post-rupture stage is significantly damaged. With the increase in confining pressure, the post-peak set is extended, and the trend of brittle to ductility transformation appears.

### 3.3. Coulomb Strength Guidelines

Figure 5 and Figure 6 show the Mohr–Coulomb envelope of coal and rock samples; the Mohr–Coulomb criterion [25] can also react in the form of failure of coal rock masses on the other hand, as shown in Equation (1):(1)τ=σtanφ+C
where *τ* is the shear stress on the shear plane; C is the cohesion of coal samples; *φ* is the internal friction angle; σ_1_ is the principal axial stress; and σ_3_ is the circumferential stress. Taking the shear stress *τ* as the ordinate and the main stress σ as the abscissa, the Mohr stress circle of the coal sample under different confining pressures is drawn. Finally, the tangent of all Mohr stress circles is made to obtain the Mohr–Coulomb strength curve of the coal sample, the angle between the strength curve and the abscissa is the friction angle (*φ*) in the coal sample. The ordinate intercept is the cohesion of the coal sample (C). The strength of coal rock under different confining pressure conditions is drawn to make Mohr’s circle as follows:

Through the drawn Mohr circle, it can be seen that, given the cohesion force of the characteristic coal sample C = 3.6, the internal friction angle *φ* = 41°, the cohesion of the typical rock sample C = 14.5, and the internal friction angle *φ* = 52°, the failure form of the specimen in the triaxial compression test is mainly the shear failure of the mono-slope surface, and the internal friction angle and cohesion of the rock sample are much greater than that of the coal sample.

### 3.4. Analysis of Failure Mode of Bearing Coal Rock Samples 

Figure 7 shows the uniaxial compression failure pattern of the coal rock mass. The coal sample is mainly brittle fracture in the uniaxial compression state, and the failure surface is irregular. After the uniaxial compression failure of coal rock, the shape of the crushed body becomes many smaller pieces, and the sizes are different. The degree of coal rock crushing is greater; the coal rock is directly crushed, there are many cracks through the coal rock, and the shots are all over the coal rock specimen. In the process of compression of fine sandstone, it is subjected to axial stress; the strength of fine sandstone is higher than that of a coal rock sample, and the boundary part of the rock sample first reaches the ultimate bearing capacity and brittle failure. After the loss of the rock sample, the fragmentation volume is more significant, and the main crack runs through the rock sample specimen. The stress drops quickly, and the residual stress remains. Different coal rocks showed brittle failure in uniaxial compression.

As shown in Figure 8, the three-axis compression failure pattern of coal rock is the coal; during the triaxial compression process of the two groups of coal rock samples, with the change of confining pressure, both groups of coal rocks showed the effect of determining pressure. The plastic characteristics of coal rock were more obvious under confining pressure. During the whole process of confining pressure change, the brittleness of coal rock weakens, and its brittleness is transformed into plasticity; the larger the confining pressure, the stronger the plasticity. Most of the triaxial compression failure locations of coal rock occur on soft structural surfaces, and only one obvious crack runs through the whole of the coal rock specimens, and other fine cracks are not obvious. The failure form of fine sandstone is mainly splitting failure and change of confining pressure. Its brittleness continues to develop into elastoplasticity; an apparent crack in the fine sandstone specimen separates the specimen, and other fine cracks are invisible to the naked eye. The failure forms of the two kinds of coal rock specimens are different under different confining pressures. The coal rock specimens have more failure forms, showing more substantial ductility and plasticity. In contrast, fine sandstone shows elastoplastic or plasticity. The failure of the two different coal rocks is mainly due to the difference in the cracks of the coal rock and the lumpiness of the broken body.

The uniaxial and triaxial failure of coal rock is also different; under the action of confining pressure, there is a primary failure surface of coal rock, and the crushed body is triangular; the larger the confining pressure, the lower the degree of crushing of the coal rock; the body is less broken, the overall integrity of the coal rock is greater, and it shows more significant ductile fracture characteristics. The single and triaxial compression of fine sandstone specimens results in brittle failures. In a confining pressure environment, the fine sandstone crushed body gradually decreases, the whole is relatively complete, showing brittle failure, and the specimen splits from the middle, manifested as a single-slope shear failure.

## 4. Characteristic Analysis of Coal Rock Characteristic Strength Parameter Response

### 4.1. Definition and Description of the Character Strength of Bearing Coal Rock Samples

As shown in Figure 9, the schematic diagram of each characteristic point in the stress–strain curve of coal rock mass is indicated; in the indoor test part of rock mechanics, the relevant typical strength parameters are widely studied, mainly through the analysis of characteristic points in the stress–strain curve, mainly including peak strength, peak strain, compaction strength, plastic strength, and residual strength. The relationship between the typical intensity parameters and the confining pressure was analyzed to analyze the response characteristics of the coal rock mass’s characteristic strength parameters. The following is a brief description of the selected feature intensity parameter definitions.

(1)Compaction (elastic) strength point of bearing coal rock sample: Due to the existence of natural joint fractures in the coal rock mass, when the axial stress is at a low range, the primary fracture is compressed and closed, and the stress–strain curve of the whole process has prominent internal concave segments. Additionally, because coal rock is a typical elastoplastic body, many primary fractures enter the elastic deformation stage after closing. It can also be called the starting point of the elastic deformation stage.(2)The Plastic strength point of bearing coal rock sample: as the stress continues to increase, the elastic deformation stage and the stress–strain have a typical linear relationship; as mentioned earlier on the elastic-plastic characteristics of the bearing coal rock mass, when the stress exceeds the plastic point, the curve will slowly begin to move away from the original linear trend and then continue to form irreversible damage accumulation until the peak strength is reached. Therefore, the starting point of the initial deviation from the linear phase is used as the plastic strength point of the bearing coal rock specimen. It can also be called the end point of the elastic deformation stage.(3)The peak intensity point of the bearing coal rock sample: This is the point corresponding to the maximum stress value in the stress–strain curve, and the axial deformation corresponding to this point is the peak strain.(4)The residual strength point of the bearing coal rock sample: The bearing coal rock sample has typical brittle characteristics. The progress of servo control technology of rigid testing machines can ensure that the post-peak curve can be obtained during the rock mechanics test. During post-peak unloading, the damage increases, but the load-bearing coal rock specimen still has a load-bearing structure, so the smallest strength value in the curve corresponds to the residual strength point.

### 4.2. Characteristic Strength Evolution Characteristics of Bearing Coal Rock Samples

The relationship between elastic modulus and confining pressure of coal rock is shown in Figure 10. Combined with theoretical analysis, the four-parameter indexes of peak intensity, peak strain, elastic modulus, and residual strength were comprehensively selected, critical data were picked up, and mathematical methods were used to fit their functional relationship with confining pressure. Then, from the perspective of macroscopic mechanics, the differences in the mechanical responses of bearer coal rock samples under different confining pressures were quantitatively studied. The specific analysis process is as follows:(1)Analysis of elastic modulus and confining pressure of bearer coal rock samples

The modulus of elasticity is one of the indicators of downhole production safety, which provides a reference for underground mining and other aspects. To further explore the stress–strain law under the confining pressure effect of different coal rocks, it is necessary to study the elastic modulus and peak strain of different coal rocks. Different rock masses have other flexible modules; for example, the elastic modulus range of fine sandstone is 27~47 GPa, and the elastic modulus of coarse sandstone is 16~40 Gpa. In this experiment, the stress–strain curve flexible stage values of different coal rock samples were selected for analysis. Under the action of confining pressure, the pore structure of coal rock quickly reaches the compacting state. With the increase in confining pressure, the coal rock becomes dense; the strength increases, and the elastic modulus increases. According to the information in Table 1 and Table 2 above, in the coal rock specimen group, the elastic modulus of coal rock is about 1.633 GPa during uniaxial compression; when the confining pressure is from 0 MPa to 3 MPa, the elastic modulus increases by about 0.2 GPa; when confining pressure is from 3 MPa to 6 MPa, it increases by about 0.15 GPa; and when confining pressure is from 6 MPa to 9 MPa, it increases by about 0.9 GPa. With the increase in confining pressure, the addition of elastic modulus slowly increases, and the effect of determining pressure on the elastic modulus of coal rock specimens begins to grow. In the fine sandstone specimen group, the elastic modulus of fine sandstone is about 11.113 GPa during uniaxial compression; when the confining pressure is from 0 MPa to 3 MPa, the elastic modulus increases by about 0.03 GPa; when the confining pressure is from 3 MPa to 6 MPa, the elastic modulus increases by about 1.4 GPa; when the confining pressure is from 6 MPa to 9 MPa, the growth is about 2.9 GPa. Under the action of confining pressure, the effect on fine sandstone is noticeable, and the elastic modulus has been dramatically increased. The development of determining pressure on coal rock specimens has been enhanced with the change of size and different confining pressure stages of coal rock specimens. The effect of confining pressure is related to the coal rock properties of the coal rock itself. The elasticity of coal rock is far less than the elasticity of fine sandstone, and the elastic properties of coal rock change significantly under the action of confining pressure; elasticity is gradually converted into plasticity, and the elasticity of fine sandstone will be enhanced with the increase in confining pressure. 

(2)Analysis of peak strength and confining pressure of bearing coal rock samples

The change curve of the peak strength of coal rock under different confining pressures is shown in Figure 11. Coal rock has a very complex structure; most scholars regard coal rock as a porous medium with fractures and believe that coal rock is mainly composed of a fractured pore–fracture system. Under the action of confining pressure, the internal pore structure of coal rock specimens has undergone tremendous changes, making the internal pore structure of coal rock denser and improving the bearing capacity. Under the three-axis compression state, the coal rock is compacted with the change of confining pressure, and its peak strength slowly increases. With the shift of confining pressure, the peak intensity of coal rock specimens changes faster and has a linear relationship with determining pressure (R^2^ = 0.98906). The peak strength of fine sandstone changes at a rate less than that of coal rock specimens and has a linear relationship with the confining pressure (R^2^ = 0.92636). Because the fine sandstone specimen itself is dense, it is a more homogeneous material, and the rate of change with confining pressure is lower than that of the coal rock specimen. At the same time, fine sandstone’s properties make its bearing capacity stronger than coal rock specimens. Under the action of confining pressure, the porous structure of coal rock is easier to change, its bearing ability is easier to adjust with confining pressure, and the peak strength can be improved more.

(3)Analysis of peak strain and confining pressure of bearing coal rock samples

Figure 12 shows the relationship between the elastic modulus of coal rock and the confining pressure. In the triaxial compression state of the coal rock sample, the force of the specimen is axial compression, resulting in axial strain. The nature of the coal rock sample itself is different, and the occurrence of its strain is also dissimilar. Due to the dense characteristics of the fine sandstone specimen, its peak strain and confining pressure have a concave nonlinear relationship (R^2^ = 0.98662). In the confinement state of 0 MPa to 3 MPa, with the increase in confining pressure, its peak strain increases; after the 3 MPa confining pressure state, its peak strain changes slowly with confining pressure and then gradually tends to stabilize, and the axial displacement is unchanged. For coal rock in the confining pressure environment, its properties change significantly; its peak strain and confining pressure have a linear relationship (R^2^ = 0.95624). The larger the confining pressure, the greater the peak strain; the axial displacement of coal rock in the process of increasing confining pressure changes significantly. The medium porous properties of coal rock are changed, and it is gradually compressed to a dense state; in the compression process, the axial displacement of coal rock changes with the confining pressure, the peak strain increases, and the effects of confining pressure on the peak strain of coal rock are more prominent.

## 5. Evolving Law of Proportion of Bearing Coal Rock Stage

### 5.1. Analysis of the Proportion of Bearing Coal Rock Stage

Table 4 and Table 5 show the statistics of the percentage of bearing coal rock sample stages of the peritectic pressure effect, respectively; combined with the selection rules of feature points, the proportion of location in the feature point area was counted separately, and the ratio of the coal-carrying stage considering the confining pressure effect was analyzed.

Figure 13 and Figure 14 show the stress–strain curve and the characteristic stage histogram of the coal rock body under different confining pressure conditions; the usual points of coal rock at each stage will change significantly with the change of confining pressure. By comparison, for the end point of the compaction stage of the coal rock sample, with the increase in confining pressure, the stress corresponding to the compaction end point continues to increase. With the addition of determining pressure, the compaction stage in the proportion of the stress curve continues to decrease, and the confining pressure begins to slowly weaken after exceeding 3 MPa, indicating that the confining pressure will affect the compaction stage of the coal rock, and the confining pressure will also make the coal rock enter the compacting state. In the elastic deformation stage, with the increase in confining pressure, the elastic strain increases continuously in the stress–strain curve, but the proportion of the elastic deformation stage decreases constantly in the whole stress–strain process. The stress value of the characteristic points corresponding to the elastic deformation continues to increase, and the proportion of the flexible stage under different confining pressure conditions is almost linear. In the yield stage, with the increase in confining pressure, the yield stage is more prominent, the characteristic point stress value of the yield stage continues to increase, and the proportion of the yield stage in the whole compression process continues to grow, indicating that with the increase in confining pressure, the brittleness of coal rock will be changed. The bearing capacity of coal rock will be increased.

### 5.2. Strength Analysis of the Evolution Stage of Bearing Coal Rock

(1)Strength analysis of coal rock compaction stage

Figure 15 shows the characteristic strength of the compacted stage of the coal rock body under different surrounding pressure states with the change of confining pressure; the power of the coal rock specimen at each location will change significantly with the shift of confining pressure, and the corresponding strength will also change. Figure 15 shows that the compaction stage strength of the coal rock specimen is less than the strength of the delicate sandstone compaction stage, and the compaction stage strength decreases with the increase in confining pressure. The change of the power of the two coal rocks is linear with the confining pressure, and the slope of the linear relationship of the compacting stage strength of the coal rock specimen is smaller than that of the fine sandstone specimen. Compared with the coal rock specimens, the compaction stage strength of fine sandstone changes faster with the shift of confining pressure, and the confining pressure effect has a noticeable influence on it. The confining result of the compacting stage of coal rock can be displayed, and the performance is not evident under high confining pressure. The degree of change of fine sandstone is more significant with the evolution of confining pressure. The reason is that the coal rock itself has low strength, and porous factors make it more susceptible to the influence of confining pressure; with the increase in confining pressure, the pore structure of coal rock changes. In the compaction stage, high confining pressure makes the coal rock begin to become dense, and the compaction stage continues to decrease. Fine sandstone has a dense structure, and the influence of confining pressure on its network is small; the change effect on its compaction stage is weaker than that of coal rock, so the strength of the compaction stage is always higher than that of coal rock.

(2)Strength analysis of the plastic stage of coal rock

Figure 16 shows the characteristic strength of the plastic stage of coal rock under different surrounding pressure states. After a period of compression, the pore structure of coal rock changes, and in the plastic background, the strength of the plastic stage will change with the shift of confining pressure. Figure 15 shows that the plastic stage strength of the coal rock specimen is less than that of the delicate sandstone plastic stage. The plastic stage strength increases with the increase in confining pressure, and the change in plastic power of the two coal rocks has a linear relationship with the confining pressure. Compared with the coal rock specimen, the plastic stage strength of acceptable sandstone changes with the confining pressure, and the rate of change is lower than that of the coal rock specimen. The confining pressure effect is more evident for the coal rock, and the degree of modification of coal rock is more significant with the change of confining pressure.

(3)Analysis of residual strength of bearing coal rock samples

Figure 17 shows the characteristic strength of coal rock residual strength under different circumferential pressure states. The stress of fine sandstone gradually decreases after rupture and finally drops close to zero, at which point the stress at almost zero can be used as the residual strength of fine sandstone. The relationship with confining pressure is plotted according to the residual power of coal rock in Table 3. It can be seen that the residual strength of fine sandstone has a nonlinear relationship with confining pressure, and the residual strength of coal rock has a linear relationship with determining pressure. There is little difference in the residual strength of coal rock samples in a uniaxial state. Under the conditions of confining pressure at 3 MPa and 6 MPa, the residual power of the rock samples increased significantly, and under the condition of confining pressure at 9 MPa, the increase was slight. The coal rock increased linearly, and the residual strength of the two types of coal rock showed different confining pressure effects. With the addition of confining pressure, the brittleness of the two kinds of coal rocks gradually transforms into plasticity. The brittleness of fine sandstone changes more significantly with confining pressure, and the residual strength changes to a large degree and shows a nonlinear change.

## 6. Discussion

The strength of the coal rock mass has a great relationship with the original pore structure of the coal rock sample, and these actual pores have an important influence on the strength of the coal rock mass. The change in the pore structure of coal rock causes the confining pressure effect of coal rock. The inside of the coal sample is mainly composed of pores with a small diameter, and the knowledge of the rock sample is primarily composed of pores with a large diameter. Under different confining pressure conditions, the tiny-diameter pores of the coal sample are extruded. The larger the confining pressure, the denser the pore structure of the coal sample, the higher the strength, and the more pronounced the confining pressure effect. The internal pore diameter of the rock sample is significant. Under the action of confining pressure, the extrusion deformation of the large diameter pore is small, which has little effect on the pore structure of the rock sample. Under confining pressure, the large-diameter pores of the coal rock mass are first extruded, and then the small-diameter pores are extruded. After some time, the internal pore structure of the specimen tends to be stable.

The pore compaction stage of coal rock mass is the large- and small-diameter pore compaction stage. In this compaction process, the coal sample pore compaction stage accounts for more than the rock sample specimen, the coal sample takes longer to stabilize the internal pore structure, and the rock sample can quickly stabilize into the next stage. In the compaction stage, the confining effect of rock samples is not apparent. The microscopic level can further reveal the confining pressure effect of the coal rock mass.

## 7. Conclusions

(1)The total stress–strain curve of coal rock under the same confining pressure conditions can be divided into four stages, namely, the four steps after pore fracture compaction, elastic deformation, yield, and rupture. The uniaxial compression is the same as the triaxial compression process. The various stages of coal rock show different changes, and the strength of the two coal rocks is more robust under the action of confining pressure, which enhances the bearing capacity of coal rock;(2)Under the conditions of 0 MPa, 3 MPa, 6 MPa, and 9 MPa, the proportion of the two groups of coal rock specimens in the pore fracture compaction stage continued to decrease, which was negatively correlated with the confining pressure. The elastic deformation stage increases in proportion through the whole process, which is positively correlated with the confining pressure. Plastic deformation positively correlates with confining pressure, which increases with increased confining pressure. The confining pressure increased, and the two coal rocks were transformed from the initial brittle characteristics to plastic;(3)Due to the pore structure of the coal rock itself, with the increase in confining pressure, the strength of the compaction stage of the coal rock continues to decrease, and the elastic modulus of the coal rock sample increases. The elastic modulus of fine sandstone is greater than that of the coal rock specimen; the degree of change of the coal rock specimen with the confining pressure is greater, and the mechanical properties of the coal rock in this process are very different. As the confining pressure increases, the peak strength of the coal rock sample increases, the residual power increases, and the effects of confining pressure on the mechanical characteristics of the coal rock specimen at each stage are more prominent.

## Figures and Tables

**Figure 1 materials-16-03235-f001:**
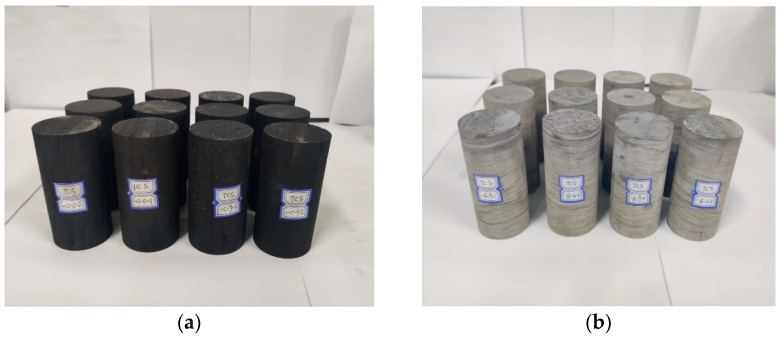
Photographs of coal rock specimens: (**a**) coal sample specimens; (**b**) rock sample specimens.

**Figure 2 materials-16-03235-f002:**
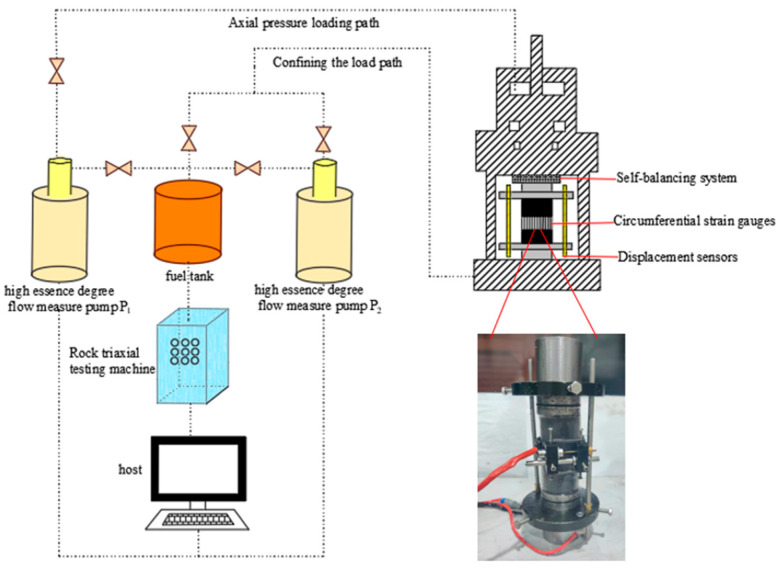
SAS-2000 multi-field-coupled rock mass dynamic disturbance triaxial theological experimental system.

**Figure 3 materials-16-03235-f003:**
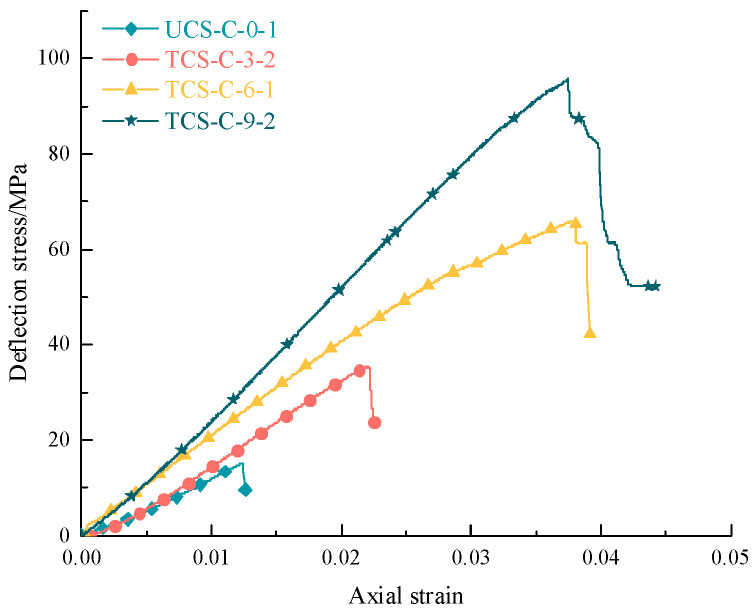
Axial stress–strain curves of typical bearing coal samples under different confining pressures.

**Figure 4 materials-16-03235-f004:**
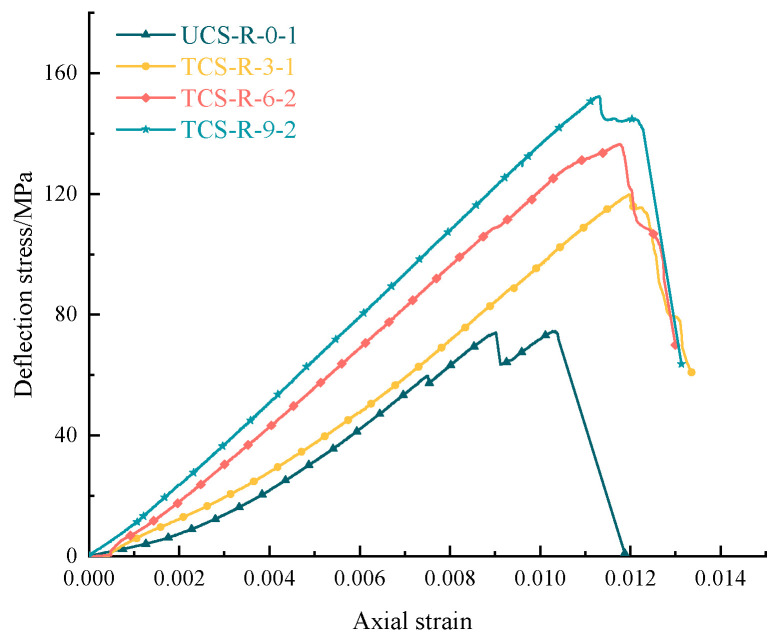
Axial stress–strain curves of typical bearing rock samples under different confining pressures.

**Figure 5 materials-16-03235-f005:**
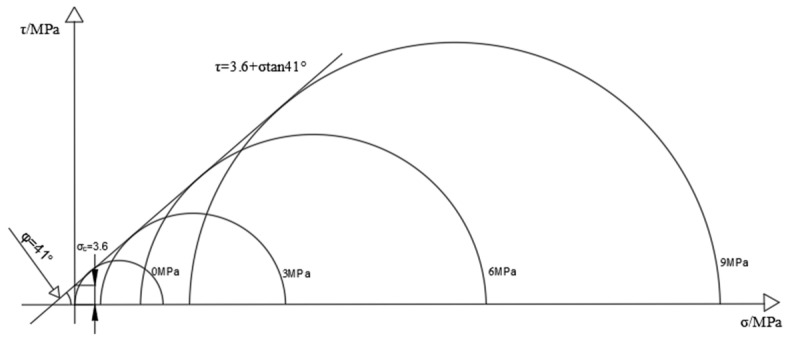
Mohr stress circle of coal sample.

**Figure 6 materials-16-03235-f006:**
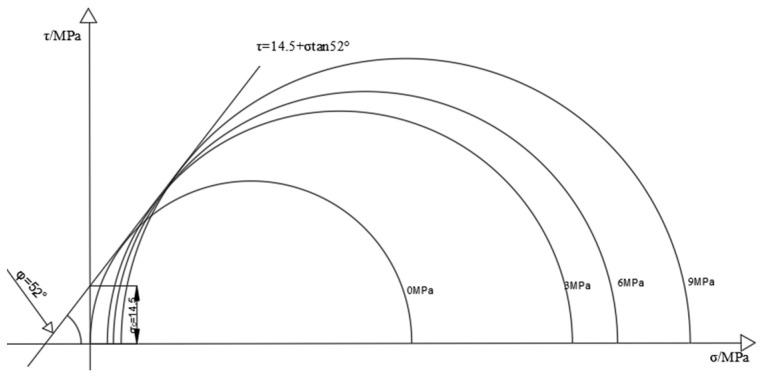
Mohr stress circle of the rock sample.

**Figure 7 materials-16-03235-f007:**
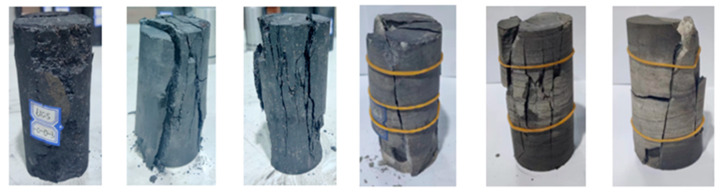
Typical uniaxial compression stress–strain curve.

**Figure 8 materials-16-03235-f008:**
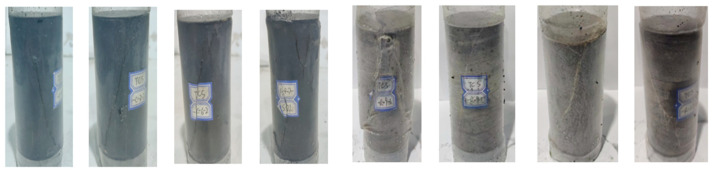
Failure pattern diagram of coal and rock under triaxial compression.

**Figure 9 materials-16-03235-f009:**
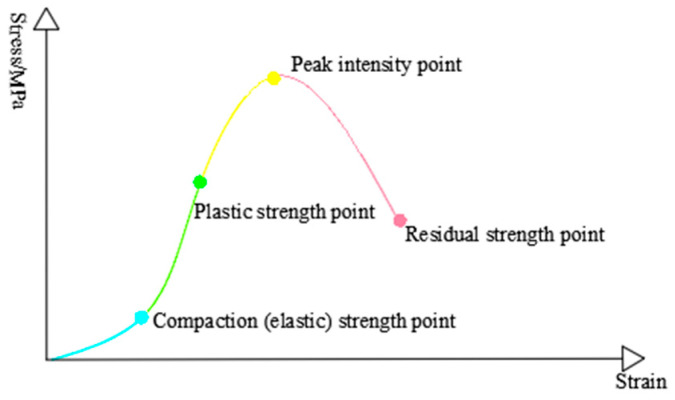
Coal rock characteristic strength point.

**Figure 10 materials-16-03235-f010:**
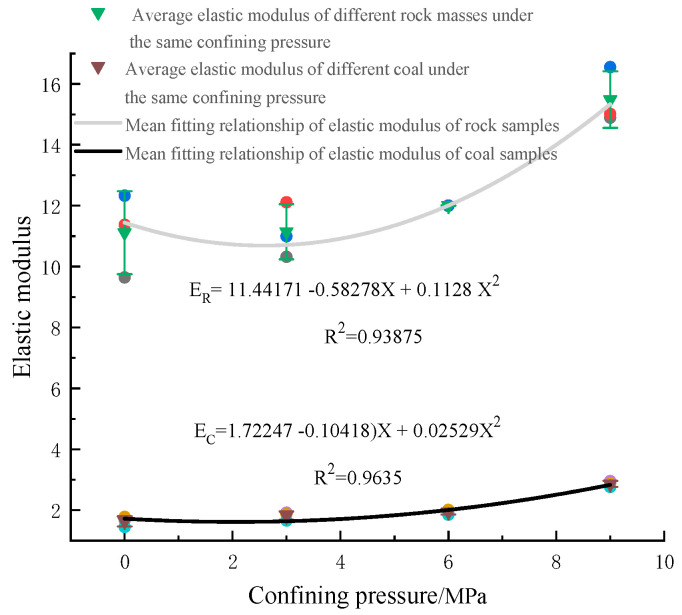
Relationship between elastic modulus of coal and rock and confining pressure. Red, blue and black are the elastic modulus at the end of the rock sample numbers “1”, “2” and “3” in Table 1 and Table 2, respectively. Yellow, purple and sky blue are the elastic modulus at the end of the coal sample numbers “1”, “2” and “3” Table 1 and Table 2, respectively. Violet, yellow, and sky blue are the peak intensities at the end of the rock sample numbers “1”, “2” and “3” in Table 1 and Table 2, respectively. Red, black, and blue are the peak intensities at the end of the coal sample numbers “1”, “2” and “3” in Table 1 and Table 2, respectively.

**Figure 11 materials-16-03235-f011:**
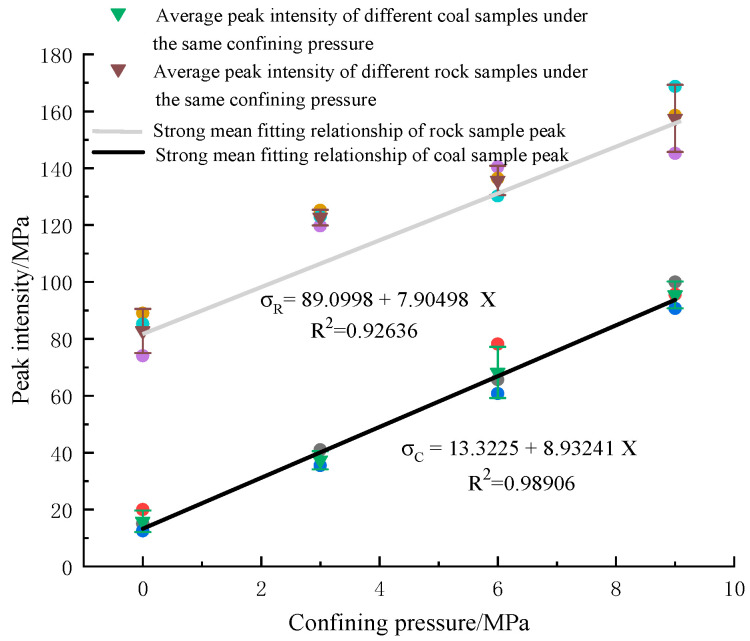
The relationship between the peak strength of coal rock and confining pressure. Violet, light blue, and orange are the peak intensities of the specimen numbers ending in “1”, “2” and “3” in Table 1 and Table 2, respectively. Blue, red, and dark blue are the peak intensities of coal samples ending in Table 1 and Table 2 ending in “1”, “2” and “3”.

**Figure 12 materials-16-03235-f012:**
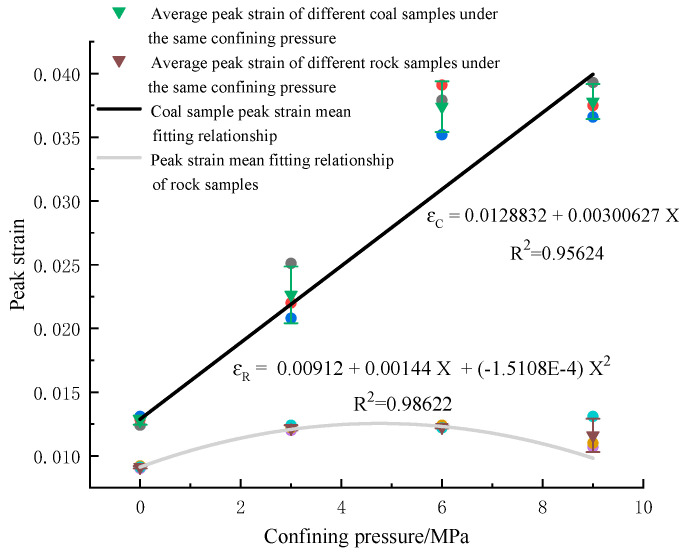
Relationship between peak strain and confining pressure in coal rock. Red, dark blue, and black are the peak strains at the end of the coal sample numbers “1”, “2” and “3” in Table 1 and Table 2, respectively. Purple, orange, and light blue are the peak strains at the end of specimen numbers “1”, “2” and “3” in Table 1 and Table 2, respectively.

**Figure 13 materials-16-03235-f013:**
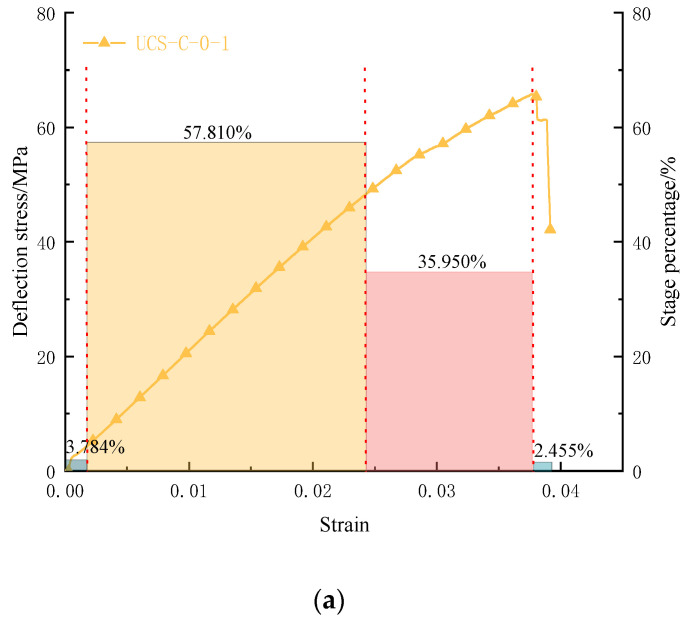
Variation law of characteristic points of coal samples during the bearing process under different confining pressures: (**a**) the proportion of characteristic intensity stages of typical bearing coal samples; (**b**) the statistical chart of the proportion of coal samples under different confining pressure conditions.

**Figure 14 materials-16-03235-f014:**
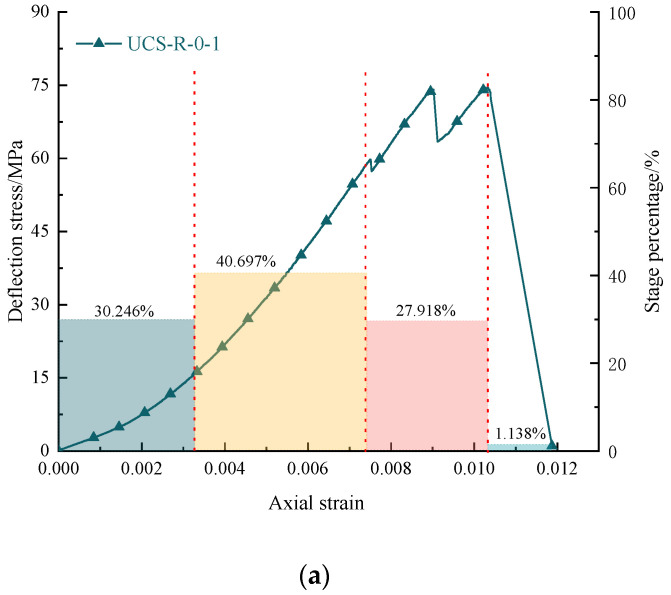
Variation law of characteristic points of rock samples during the bearing process under different confining pressures: (**a**) the proportion of the characteristic strength stage of the typical bearing rock sample; (**b**) statistical chart of the proportion of bearing rock sample stages under different confining pressure conditions.

**Figure 15 materials-16-03235-f015:**
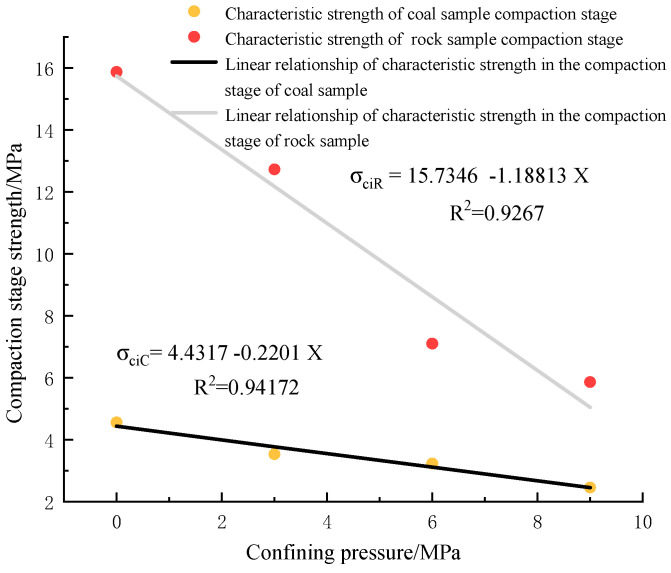
Characteristic strength of coal rock compaction stage under different confining pressure conditions.

**Figure 16 materials-16-03235-f016:**
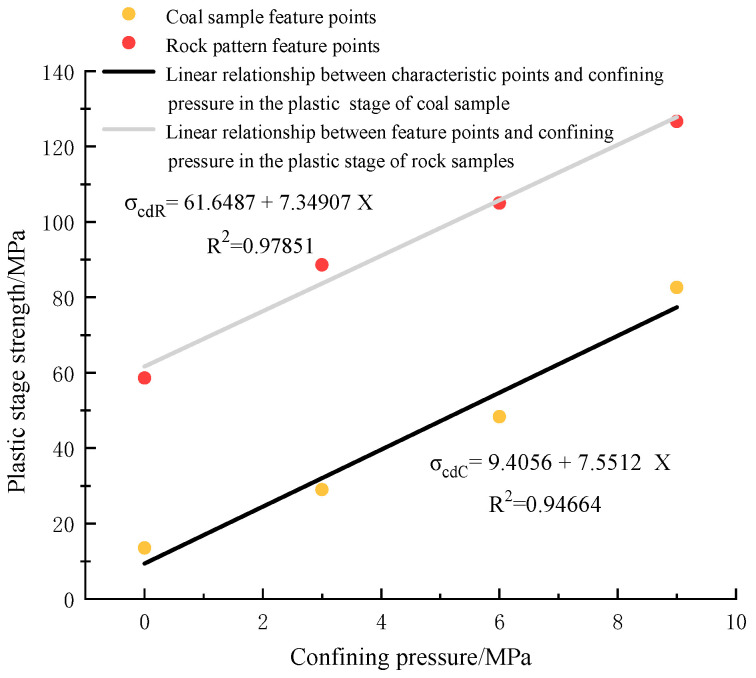
Characteristic strength of plastic stage of coal rock under different confining pressure conditions.

**Figure 17 materials-16-03235-f017:**
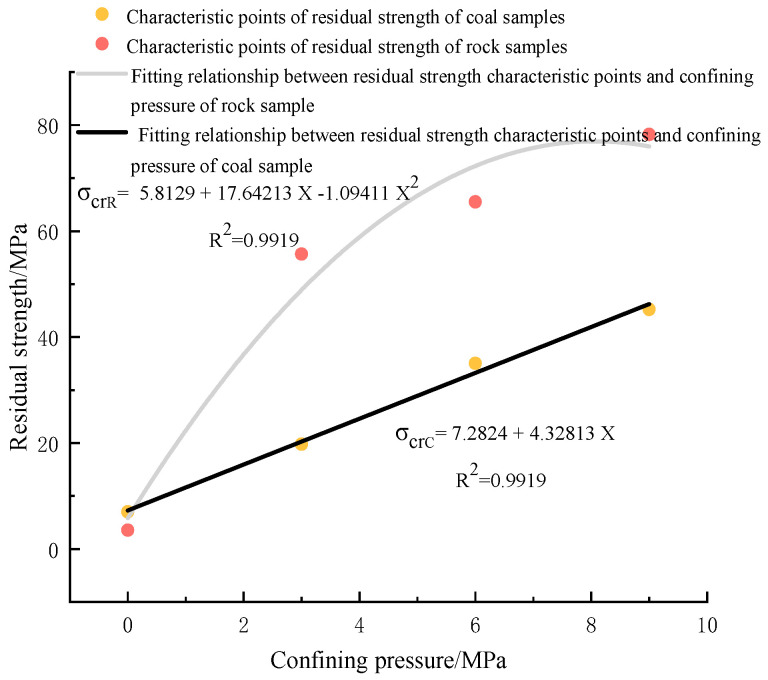
Characteristic strength of coal rock residual strength under different confining pressure conditions.

**Table 1 materials-16-03235-t001:** Statistical table of basic mechanical parameters of coal samples considering the confining pressure effect.

Specimen Number	Peak Intensity/MPa	Peak Strain	Young’s Modulus/GPa
UCS-C-0-1	15.074	0.0124	1.670
UCS-C-0-2	20.021	0.0129	1.780
UCS-C-0-3	12.516	0.0131	1.450
Average value	15.870	0.0128	1.633
TCS-C-3-1	41.038	0.0251	1.921
TCS-C-3-2	35.480	0.0220	1.870
TCS-C-3-3	32.553	0.0208	1.661
Average value	36.357	0.0226	1.817
TCS-C-6-1	65.689	0.0379	1.970
TCS-C-6-2	78.220	0.0391	2.010
TCS-C-6-3	60.781	0.0368	1.852
Average value	68.230	0.0379	1.944
TCS-C-9-1	100.031	0.0393	2.957
TCS-C-9-2	95.681	0.0375	2.870
TCS-C-9-3	90.664	0.0419	2.763
Average value	95.459	0.0396	2.863

**Table 2 materials-16-03235-t002:** Statistical table of mechanical parameters of rock sample foundation considering the confining pressure effect.

Specimen Number	Peak Intensity/MPa	Peak Strain	Young’s Modulus/GPa
UCS-R-0-1	74.099	0.0090	9.6400
UCS-R-0-2	89.065	0.0092	11.370
UCS-R-0-3	85.226	0.0091	12.330
Average value	82.797	0.0091	11.113
TCS-R-3-1	119.715	0.0120	10.320
TCS-R-3-2	125.220	0.0122	12.112
TCS-R-3-3	123.095	0.0124	10.996
Average value	122.677	0.0122	11.143
TCS-R-6-1	140.481	0.0122	12.011
TCS-R-6-2	136.438	0.0118	13.670
TCS-R-6-3	130.226	0.0108	11.995
Average value	135.715	0.0116	12.559
TCS-R-9-1	145.209	0.0107	14.891
TCS-R-9-2	152.279	0.0113	13.481
TCS-R-9-3	168.750	0.0127	16.553
Average value	155.413	0.0116	14.975

**Table 3 materials-16-03235-t003:** Statistical table of residual strength parameters of coal and rock samples considering the confining pressure effect.

Specimen Number	Residual Strength/MPa	Specimen Number	Residual Strength/MPa
UCS-C-0-1	9.558	UCS-R-0-1	1.025
UCS-C-0-2	8.920	UCS-R-0-2	4.181
UCS-C-0-3	7.021	UCS-R-0-3	3.556
Average value	8.500	Average value	2.921
TCS-C-3-1	20.066	TCS-R-3-1	60.901
TCS-C-3-2	23.652	TCS-R-3-2	50.025
TCS-C-3-3	19.764	TCS-R-3-3	55.663
Average value	21.161	Average value	55.530
TCS-C-6-1	42.143	TCS-R-6-1	74.403
TCS-C-6-2	39.668	TCS-R-6-2	70.005
TCS-C-6-3	35.041	TCS-R-6-3	65.507
Average value	38.951	Average value	69.972
TCS-C-9-1	47.005	TCS-R-9-1	80.602
TCS-C-9-2	52.279	TCS-R-9-2	75.708
TCS-C-9-3	45.210	TCS-R-9-3	78.226
Average value	48.165	Average value	78.155

**Table 4 materials-16-03235-t004:** Statistics on the proportion of bearing coal sample stage considering the confining pressure effect.

Confining Pressure/MPa	Compaction Stage Percentage/%	Elastic Stage Percentage/%	The Proportion of PlasticStage/%	Post-Rupture Stage/%
0	27.549	59.236	11.644	1.570
3	13.806	66.313	19.096	0.785
6	8.784	70.810	16.950	4.455
9	3.498	73.088	12.583	9.352

**Table 5 materials-16-03235-t005:** Statistics on the bearing rock sample stage proportion considering the confining pressure effect.

Confining Pressure/MPa	Compaction Stage Percentage/%	Elastic Stage Percentage/%	The Proportion of PlasticStage/%	Post-Rupture Stage/%
0	30.246	40.697	27.918	1.138
3	12.753	60.941	21.664	4.643
6	4.762	66.344	26.477	2.453
9	2.517	71.719	9.735	16.029

## Data Availability

The data presented in this study are available on request from the corresponding author.

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
