# Peer review of "Research on the Response Mechanism of Coal Rock Mass under Stress and Pressure"

_materials, 2023, doi:10.3390/ma16083235_

Round 1
Reviewer 1 Report
This manuscript suffers from serious flaws in terms of both language and content. Grammar is often very poor, verbs are missing, words are used wrongly and I was seriously in trouble trying to understand the sentences.
Moreover, the content is weak: although the laboratory testing campaign is extensive, the results are not properly processed. A wide discussion on the behaviour of the materials should be strongly based on a complete evaluation of the mechanical properties, not only on random comparisons of results. Also, the empirical fitting laws shown in figures 8, 9, 10 are not based on or compared with similar trends in the literature, nor the discussion of the results is somehow referred to previous works in the literature.
Some specific comments are listed in the following.
- keywords: destruction is not a technical term, please correct
- lines 34-37: the sentence is totally unclear and too long, please rewrite it.
- line 37: In which sense you talk about force? Probably you mean the resistance. Please explain better.
- line 39: You probably mean "obtain". Please check the sentence.
- line 42: what do you mean with occurrence state? please explain.
- line 45: Totally unclear. Explain what you mean in a better way.
- lines 48-49: totally unclear, please rewrite
- line 82: unclear, as before
- line 87-91: sentence too long and completely unclear, rewrite
- lines 99-105: totally unclear, explain better
- line 109: specify that they are the radius and height of the specimen
- section 2.2: Grammar is very poor, many verbs are missing and sentences are totally unclear.
- line 155: in which sense "false"?
- section 3: You should add the failure envelope and derive Mohr Coulomb parameters. This is fundamental to characterize the material behaviour in a standard and comparable way.
- figures 5 and 6: there is no need to add the drawings below the images
- figure 8: add legend with the meaning of colors (same comment for the following figures 9 and 10)
- References should not be made only to chinese authors
Reviewer 2 Report
The manuscript concerns the study of the strength and deformation characteristics of coal samples and the surrounding rock mass. The authors performed laboratory tests on coal and rock samples in both uniaxial and triaxial conditions at various confining pressures. Then, the results were analyzed in terms of mechanical properties (compressive strength, deformation, Young's modulus). The analyzes also included the analysis of the course of stress/strain curves for the laboratory tests carried out and the determination of the nature of the damage to the samples as well as statistical analyses. The article seems to be interesting and necessary for scientific and engineering reasons. In my opinion, it is fit for publication after making corrections. My comments are:
1. Markings should be adapted to general standards, e.g. φ should be replaced with Ï•;
2. Please change Deflection stress to Stress or Pressure;
3. Figure 12(b) should be improved because of its low quality;
4. The literature should be supplemented with world achievements in this field. The authors rely only on the works of Chinese authors. This needs to be corrected.
Reviewer 3 Report
This is an interesting study on the strength characteristics and deformation of coal under confining stress with great relevance for coal mining. The methodology and presentation of results are straightforward and properly follow design of experiments. This reviewer has only a few suggestions before publication.
1. The title is not clear and may be difficult for readers to understand. This reviewer sugests altering to "Research on the response mechanism of coal rock mass under stress and pressure" or "Investigating the co-evolution of stress and structure in coal rock mass".
2. Please introduce all figures and tables in the text before they appear in the manuscript.
3. The results and findings should be compared to and discussed in the context of earlier works cited in the literature review
Round 2
Reviewer 1 Report
The quality of the manuscript improved a lot, but still it is not clear which is the aim of the research. Basically, you test rock and coal, considering possible collapse in coal mines, but you never try to understand the possible consequences of such different mechanical and deformation properties of the two materials. Also, previous work are not considered as reference for comparing your results. I suggest to improve the discussion in light od this. Please avoid general terms (i.e. relatively, quite. etc.), refer always to a quantitative description.
